# Intra-Arrest Therapeutic Hypothermia and Neurologic Outcome in Patients Admitted after Out-of-Hospital Cardiac Arrest: A Post Hoc Analysis of the Princess Trial

**DOI:** 10.3390/brainsci12101374

**Published:** 2022-10-10

**Authors:** Elisabetta MACCHINI, Emelie DILLENBECK, Martin JONSSON, Filippo ANNONI, Sune FORSBERG, Jacob HOLLENBERG, Anatolij TRUHLAR, Leif SVENSSON, Per NORDBERG, Fabio Silvio TACCONE

**Affiliations:** 1Department of Intensive Care, Hôpital Universitaire de Bruxelles (HUB), 1070 Brussels, Belgium; 2Department of Clinical Science and Education, Södersjukhuset, Centre for Resuscitation Science, Karolinska Institutet, 17177 Stockholm, Sweden; 3Emergency Medical Services of the Hradec Kralove Region, Department of Anaesthesiology and Intensive Care Medicine, Charles University in Prague, University Hospital Hradec Kralove, 50022 Hradec Kralove, Czech Republic; 4Department of Medicine, Karolinska Institute, 17177 Stockholm, Sweden; 5Function Perioperative Medicine and Intensive Care, Karolinska University Hospital, 17177 Stockholm, Sweden

**Keywords:** intra-arrest hypothermia, outcome, cardiac arrest, trans-nasal evaporative cooling

## Abstract

*Background*: Despite promising results, the role of intra-arrest hypothermia in out-of-hospital cardiac arrest (OHCA) remains controversial. The aim of this study was to assess the effects of trans-nasal evaporative cooling (TNEC) during resuscitation on neurological recovery in OHCA patients admitted alive to the hospital. *Methods*: A post hoc analysis of the PRINCESS trial, including only patients admitted alive to the hospital, either assigned to TNEC or standard of care during resuscitation. The primary endpoint was favorable neurological outcome (FO) defined as a Cerebral Performance Category (CPC) of 1–2 at 90 days. The secondary outcomes were overall survival at 90 days and CPC 1 at 90 days. Subgroup analyses were performed according to the initial cardiac rhythm. *Results*: A total of 149 patients in the TNEC and 142 in the control group were included. The number of patients with CPC 1–2 at 90 days was 56/149 (37.6%) in the intervention group and 45/142 (31.7%) in the control group (*p* = 0.29). Survival and CPC 1 at 90 days was observed in 60/149 patients (40.3%) vs. 52/142 (36.6%; *p* = 0.09) and 50/149 (33.6%) vs. 35/142 (24.6%; *p* = 0.11) in the two groups. In the subgroup of patients with an initial shockable rhythm, the number of patients with CPC 1 at 90 days was 45/83 (54.2%) in the intervention group and 27/78 (34.6%) in the control group (*p* = 0.01). *Conclusions*: In this post hoc analysis of admitted OHCA patients, no statistically significant benefits of TNEC on neurological outcome at 90 days was found. In patients with initial shockable rhythm, TNEC was associated with increased full neurological recovery.

## 1. Introduction

The use of targeted temperature management (TTM) remains controversial in patients successfully resuscitated from cardiac arrest (CA) [1,2,3]; indeed, after the publication of two small randomized controlled trials (RCTs) showing an improvement in neurological recovery and survival for out-of-hospital cardiac arrest (OHCA) patients with an initial shockable rhythm treated at 32–34 °C for 12–24 h [4,5], three large RCTs have provided additional information: (a) TTM at 33 °C resulted in similar rates of neurological recovery and survival as TTM at 36 °C in OHCA, regardless of the initial rhythm [6]; (b) TTM at 33 °C resulted in a better neurological outcome than normothermia in an heterogenous population of OHCA and in-hospital CA with an initial non-shockable rhythm [7]; and (c) TTM at 33 °C results in similar outcomes as controlled normothermia in a large OHCA population [3]. While some researchers consider that TTM should not be considered an effective intervention in CA patients, others argue that TTM might be still beneficial in some specific populations, in particular when it is of “high quality” [8].

Among all characteristics that define “high quality TTM”, early exposure to cooling appears to be one of the most effective, at least in the experimental setting. In particular, animal data suggest that hypothermia initiated during cardiopulmonary resuscitation (CPR), so-called intra-arrest hypothermia (IATH), would provide more robust neuroprotective effects than cooling initiated at a later stage, after the return of spontaneous circulation (ROSC); this would be particularly relevant for some specific cooling techniques, such as total lung ventilation or trans-nasal evaporative cooling (TNEC) [9,10,11,12].

In one phase-II RCT (PRINCE), TNEC was associated with a lower body temperature on arrival and a trend towards a higher number of patients with intact neurological recovery among those admitted to the hospital after sustained ROSC [13]. In a phase-III RCT (PRINCESS), 677 unconscious OHCA patients were randomized to receive either TNEC during CPR or cooling started after hospital arrival [14]; the proportion of patients achieving good neurological outcome at 90 days was similar between the groups. In a pooled analysis of individual data from these two trials, TNEC was associated with a significant increase in favorable neurological outcome in out-of-hospital cardiac arrest patients with initial shockable rhythms [15].

The aim of this *post hoc* analysis was to assess the effects of TNEC on neurological outcome in the PRINCESS trial considering only the patients admitted alive to the hospital after ROSC.

## 2. Materials and Methods

### 2.1. Trial Design

The PRINCESS trial was a randomized clinical study with a blinded assessor of the primary outcome, which was conducted in 11 emergency medical service (EMS) systems in seven European countries between 1 January 2010 and 31 January 2018. Each participating center approved the study protocol according to national legislations; the statistical analysis plan and the rationale and design of the trial were published before the end of the enrolment phase [14]. The study was conducted according to the Declaration of Helsinki. The data were reviewed by an independent data and safety monitoring committee with two interim analyses: one after the recruitment of 200 and the second of 500 randomized patients. Following the second interim analysis, the recruitment of patients by helicopter was halted because of the long time period from collapse to inclusion and application of the intervention. Written informed consent was obtained from the family or a legal representative for each patient after hospital admission; for those patients regaining mental capacity, confirmation of the written consent was also requested.

### 2.2. Patients

The inclusion criteria of the PRINCESS trial were: (a) witnessed cardiac arrest; (b) age ≥ 18 years. The exclusion criteria were: >80 years of age; cardiac arrest secondary to trauma, severe hemorrhage, drug overdose, acute brain injury, drowning, smoke inhalation, electrocution, hanging; hypothermia (i.e., body temperature <34 °C) on EMS arrival; the presence of an anatomic barrier to place the intra-nasal catheters; do-not-resuscitate order; terminal disease; pregnancy; known coagulopathy (but not due to chronic therapy with anticoagulants); supplemental oxygen therapy at home; ROSC before randomization; collapse to EMS arrival >15 min. In this post hoc analysis, patients who died before admission to hospital were excluded from both groups.

### 2.3. Randomization and Trial Intervention

The patients were screened by the advanced life support team after initial airway management (i.e., endotracheal intubation or laryngeal mask) for eligibility. If all inclusion and exclusion criteria were fulfilled, the patients were therefore randomized (ratio of 1:1) to receive either the intervention (i.e., intra-arrest cooling) or standard care, using sequentially numbered envelopes (blocks of four) provided to each study site by the Karolinska Institute. Advanced life support care was provided in both groups according to international guidelines. For patients randomized to the intervention group, intra-arrest cooling was provided using a mixture of air or oxygen and a liquid coolant (perfluorohexane) via nasal catheters, as previously described [13,14]. Briefly, the TNEC was provided through the RhinoChill system, which consists of a control unit, disposable nasal catheters, two 1 L bottles of coolant and an oxygen/air tank, with an autonomy of nearly 30 min. After randomization, the nasal catheters were fully inserted through the nostrils into the nasal cavity, the control unit was activated and the coolant was nebulized by close contact with oxygen/air (40 L/min) at a temperature of 2–4 °C. Additional oxygen/air tanks or a connection to an ambulance or hospital oxygen supply was required for longer use in case of ROSC (with the second bottle of oxygen, cooling is possible for up one hour).

In cases of ROSC or transport to the hospital under CPR, intra-arrest cooling was continued until systemic cooling (i.e., surface blankets or intravascular catheters with temperature feedback) was initiated, after hospital admission. All admitted patients received post-resuscitation care, including systemic hypothermia at 33 °C for 24 h in both groups. Rewarming was kept between 0.2 °C to 0.5 °C per hour and fever was avoided until 72 h after randomization. All other interventions were provided according to local protocols [14].

### 2.4. Data Collection

The data on patients’ characteristics during resuscitation followed the Utstein template. The EMS team recorded prehospital event times and temperature at ROSC (whenever possible). Tympanic and core temperature, gas analyses, hemodynamics and EKG were collected after hospital arrival. Other events and measures, such as coronary angiography, intra-aortic balloon counter-pulsation (IABP), adverse events and neurologic prognostic measures, were also recorded during the ICU stay. At 90 days after randomization, the data on neurological outcome were collected using a structured phone interview or a person-to-person meeting; the Pittsburgh cerebral performance category (CPC) scale of 1 (alert and has normal cerebral function) or 2 (alert and has sufficient cerebral function to live independently and work in a sheltered environment) were considered as favorable neurological outcome. In addition, CPC 3 (i.e., severe disability; conscious, but dependent on others for daily support), CPC 4 (i.e., vegetative state; any degree of coma without the presence of all brain-death criteria) and CPC 5 (i.e., death) were considered as unfavorable neurological outcomes.

### 2.5. Outcomes

The primary outcome of this study was the occurrence of favorable neurological outcome at 90 days in the group of patients admitted alive to the hospital, according to the arm assignment. The secondary outcomes were overall survival at 90 days and CPC 1 at 90 days. Subgroup analyses of the primary and secondary endpoints were also performed in the admitted patients with an initial shockable and non-shockable rhythm. Primary outcome analyses were also adjusted for several confounders, using a random effect model including age, gender, bystander CPR and location of arrest, with the participating center as a random effect. The exploratory analyses included various baseline characteristics. The data of the current study are presented following the Strengthening the Reporting of Observational Studies in Epidemiology (STROBE) guidelines.

### 2.6. Statistical Analysis

The primary outcome trial analysis was performed in the modified intention-to-treat population, which was defined as all randomly assigned patients except those not fulfilling the inclusion criteria and never receiving the intervention. Continuous variables were reported as medians (IQRs) or mean (SD), according to their distribution; categorical variables were reported as counts and percentages. Primary analyses for main end points were conducted with chi-square tests for the comparison of binominal proportions; in addition, the odds ratio (OR) with a 95% confidence interval (CI) was computed. Additional subgroup analyses, as presented in the entire cohort of the PRINCESS study, were also performed, and a risk ratio with 95% confidence intervals (CI) was computed. All probability values were two-sided, with values less than 0.05 regarded as statistically significant. All statistical analyses were performed with R (version 3.4.3).

## 3. Results

### 3.1. Study Population

Of the 677 included patients, a total of 291 (43.0%) were admitted alive at the hospital after ROSC: 149 patients in the intervention group and 142 in the control group. The patient characteristics, factors at the scene of the arrest, resuscitation measures, and event times prior to randomization were similar in the two groups (Table 1). Patients in the intervention group experienced CA less often at home, but received CPR from bystanders less frequently than the others.

Among all patients admitted to the hospital, 9/149 (6.0%) in the intervention group and 16/142 (11.3%) in the control group had a reoccurrence of arrest in the field before persistent and sustained ROSC on admission was observed. The duration of CPR was significantly longer in the intervention group compared to the controls; moreover, in patients randomized to the intervention (i.e., median time from arrest to start of cooling was 17 min), the median tympanic temperature at ROSC was significantly lower than others (Table 2). All other characteristics on admission, as well as treatments for cardiac arrest, were similar between the groups.

### 3.2. Primary Outcome

The number of patients who survived with favorable neurologic function (CPC 1–2) at 90 days was 56 of 149 (37.6%) in the intervention group vs. 45 of 142 (31.7%) in the control group (difference, 5.9% [95% CI, −5.2% to 16.8%]; OR, 1.19 [95% CI, 0.86–1.63]; *p* = 0.29) (Table 3). In the random effects model analysis, the OR was 1.13 [95% CI, 0.63–2.07]; *p* = 0.55).

### 3.3. Secondary Outcomes

Overall survival at 90 days was observed in 60 of 149 patients (40.3%) in the intervention group vs. 52 of 142 (36.6%) in the control group (difference, 3.7% [95% CI −7.5% to 14.8%]; OR, 1.10 [95% CI, 0.82–1.47]; *p* = 0.09) (Table 4). The number of patients who survived with CPC 1 at 90 days was 50 of 149 (33.6%) in the intervention group vs. 35 of 142 (24.6%) in the control group (difference, 9.0% [95% CI −1.5% to 19.3%]; OR, 1.36 [95% CI, 0.94–1.96]; *p* = 0.11—Table 4). No significant differences between intervention and control groups were observed for primary outcome in exploratory analyses (Figure 1).

### 3.4. Subgroup Analyses

In the subgroup of patients with an initial shockable rhythm, favorable neurologic function was observed in 48/83 (57.8%) patients in the intervention group vs. 35/78 (44.9%) in the control group (difference, 12.9% [95% CI −2.4% to 28.3%]; OR, 1.29 [95% CI, 0.95–1.75]; *p* = 0.10; Table 3). Moreover, survival at 90 days was observed in 51/83 patients (61.4%) in the intervention group vs. 41/78 (52.6%) in the control group (difference, 8.8% [95% CI −6.4% to 24.1%]; OR, 1.17 [95% CI, 0.89–1.53]; *p* = 0.25). The number of patients who survived with CPC 1 at 90 days was 45 of 83 (54.2%) in the intervention group vs. 27 of 78 (34.6%) in the control group (difference, 19.6% [95% CI, 4.6–34.6%]; OR, 1.57 [95% CI, 1.09–2.25]; *p* = 0.01—Table 4). No effects of TNEC were observed in patients with non-shockable rhythms. No significant differences between the intervention and control groups were observed for primary outcome in the additional exploratory analyses (Figure 2). In the random effects model analysis, the OR for favorable neurological outcome in patients with an initial shockable or non-shockable rhythm were 1.51 [95% CI, 0.70–3.24] (*p* = 0.23) and 0.65 [95% CI, 0.33–1.99] (*p* = 0.45), respectively.

The distribution of the CPC scores between groups, according to the initial rhythm, is shown in Figure 3.

## 4. Discussion

In this study, we observed that intra-arrest cooling using TNEC did not significantly improve outcomes in admitted OHCA patients when compared to standard of care. However, in the subgroup of patients with initial shockable rhythm, we found TNEC to be associated with an increased probability of full neurological recovery.

The main findings from this post hoc analysis are in line with previous publications [14,15]; however, these findings also provide additional evidence to the existing literature. In particular, we evaluated the effects of TNEC only in admitted patients [16]. As TNEC did not impact the ROSC rate or the number of patients admitted to the hospital (i.e., no harm related to the introduction of an additional therapeutic intervention in the field together with standard CPR), it is logical to hypothesize that the neuroprotective effects from early cooling could only be observed in patients with sustained ROSC. The lack of statistically significant differences between groups might suggest that this analysis was underpowered; as such, future trials’ design and sample size calculation should consider these findings to develop an adequate cohort to test the effectiveness of TNEC in this setting. As in previous publications, most of the benefits of intra-arrest cooling was observed in patients with an initial shockable rhythm [13,14,15]. Interestingly, these data do not coincide with those from a previous randomized trial, showing an improvement in neurological outcome when TTM was applied in comatose survivors after cardiac arrest with non-shockable rhythm, when compared to targeted normothermia [7]. Additionally, recent studies showed no effects of TTM initiated after ROSC in a large cohort of OHCA patients, mostly with an initial shockable rhythm [3]. These discrepancies between our findings and previous studies might be explained by: (a) patients’ selection (i.e., only witnessed CA was selected in this analysis); (b) selection bias (i.e., we only included admitted patients, which is against the application of the “intention-to-treat” analysis proposed in large randomized studies); and (c) the potential advantage of intra-arrest cooling over standard TTM after ROSC, i.e., early initiation of hypothermia can increase its effectiveness as a neuroprotective intervention, although this analysis is exploratory and only hypothesis-generating. Future randomized trials including larger cohorts of OHCA patients with an initial shockable rhythm should be targeted to assess the expected benefits from intra-arrest cooling with the use of trans-nasal evaporative or other cooling devices. Moreover, whether these results could be generalizable among resuscitation teams that are not expert in the use of this cooling technique remains to be evaluated.

This is not the first study conducted in cardiac arrest patients evaluating the differences in patient outcomes between all randomized patients and those admitted to the hospital. As an example, intravenous adrenaline can increase the ROSC rate when compared to placebo (i.e., an increased number of patients admitted to the hospital); however, the proportion of patients with favorable neurological outcome was similar between groups [17,18]. Whether only admitted patients should be analyzed in future studies assessing the effects of intra-arrest hypothermia in cardiac arrest patients remains controversial. Importantly, this approach would still require a statistical adjustment for potential confounders, to avoid overestimation of the intervention effect due to baseline imbalances between groups. Moreover, in this study, we observed a higher proportion of patients with full neurological recovery (i.e., CPC 1) among the treated patients with initial shockable rhythm compared to others. Although CPC is not recommended as the optimal neurological scale to assess neurological recovery in this setting [19], the clinical difference between CPC 1 (i.e., conscious patient, able to work with only mild neurological or psychological deficit) and CPC 2 (i.e., conscious, independent in daily life, but only able to work in a sheltered environment), either at hospital discharge or a long-term follow-up, is important and remains consistent [20]. Differences in CPC 1 and 2 rates have been poorly reported for cardiac arrest studies. In other TTM studies, the occurrence of CPC 1 and 2 was similar between the hypothermia and the control groups [3,7]. In a small study evaluating the effects of erythropoietin in cardiac arrest patients, the CPC 1 rate was higher in the treated group (55% vs. 38%) [21]; however, these results were not confirmed in a larger randomized study [22]. Future studies should adequately report potential differences in the occurrence of CPC 1 and 2, as well as of the biomarkers of brain injury, in different study groups to better quantify the effects of several interventions on brain recovery in this setting.

This study has several limitations. First, the study cohort could be underpowered to further assess the effects of TNEC among admitted CA patients. Second, the patient outcomes were only adjusted for some relevant characteristics or confounders between groups. Third, translating these findings into the methodology of a new study remains a complex issue, as at the moment of randomization (i.e., during resuscitation), it is not possible to identify those patients eventually achieving ROSC and this would create an imbalance between the groups. Fourth, the in-hospital management of admitted patients was not entirely standardized among the centers, and post-resuscitation care might also have influenced the final results. Fifth, no additional data on the effects of the intervention on other tools to assess brain damage (i.e., serum biomarkers) were available. Finally, no assessment of the quality of CPR, a key determinant of ROSC and favorable outcome in this setting, was available in either group.

## 5. Conclusions

In this post hoc analysis of admitted patients in the PRINCESS trial, we found no statistically significant benefits in the whole population of TNEC on neurological outcome at 90 days. However, TNEC was associated with an improved complete neurological recovery (CPC1) among admitted OHCA patients with an initial shockable rhythm compared with standard TTM initiated after hospital admission. Future studies should target this patient population to confirm these findings.

## Figures and Tables

**Figure 1 brainsci-12-01374-f001:**
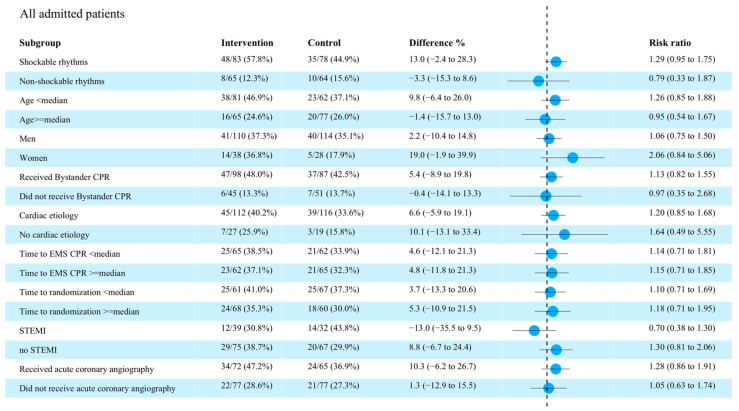
Forest plot for all patients admitted alive to hospital. CPR = cardiopulmonary resuscitation; EMS = emergency medicine system; STEMI = ST-elevation myocardial infarction.

**Figure 2 brainsci-12-01374-f002:**
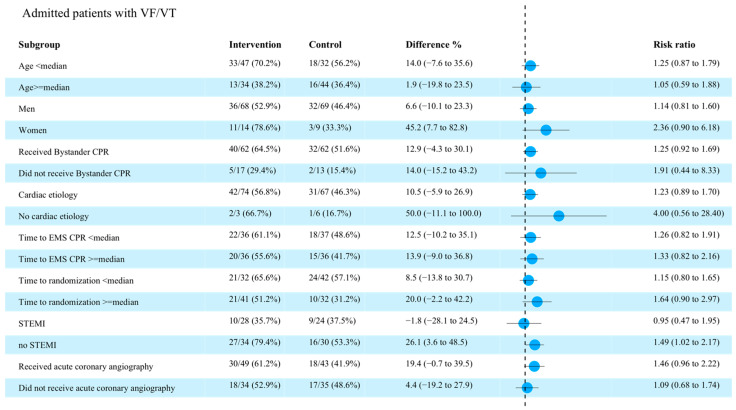
Forest plot in the subgroup of patients with an initial shockable rhythm. CPR = cardiopulmonary resuscitation; EMS = emergency medicine system; STEMI = ST-elevation myocardial infarction.

**Figure 3 brainsci-12-01374-f003:**
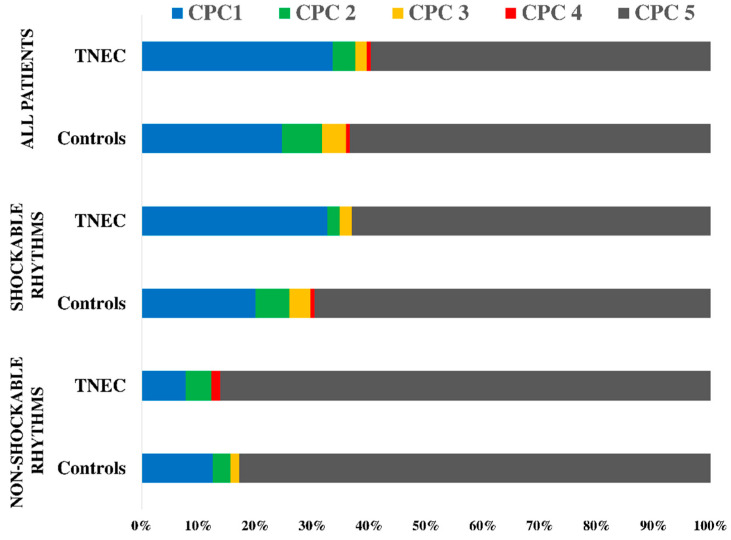
Cerebral Performance Score (CPC) at 6-month distribution between study groups in the subgroup of patients with shockable (ventricular fibrillation or pulseless ventricular tachycardia) and non-shockable (asystole and pulseless electric activity) initial rhythm.

**Table 1 brainsci-12-01374-t001:** Baseline characteristics of patients, according to the trial arm assignment. Data are reported as median [IQRs] or count (%). CPR = cardiopulmonary resuscitation; EMS = emergency medicine system; ALS = advanced life support; COPD = chronic obstructive pulmonary disease.

	Intervention (*n* = 149)	Control (*n* = 142)	*p-*Value
*Demographic characteristics*			
Age, years	62 [53–70]	65 [57–71]	0.13
Male sex, *n* (%)	110 (73.8%)	114 (80.2%)	0.23
Height, cm	177 [170–180]	178 [170–180]	0.78
Weight, Kg	81 [70–90]	85 [75–95]	0.16
*Resuscitation Characteristics, No./Total (%)*			
Location at home, *n* (%)	70 (47.0%)	86 (60.5%)	0.02
Presumed cardiac cause, *n* (%)	112 (75.2%)	116 (81.7%)	0.24
Shockable rhythm, *n* (%)	83 (55.7%)	78 (54.9%)	0.84
Bystander CPR performed, *n* (%)	98 (65.8%)	87 (61.3%)	0.33
CPR by first responder, *n* (%)	79 (53.0%)	92 (64.8%)	0.06
*Medical history*			0.33
Coronary artery disease, *n* (%)	24 (16.1%)	33 (23.2%)	
Hypertension, *n* (%)	29 (19.4%)	31 (21.8%)	
COPD, *n* (%)	9 (6.0%)	4 (2.8%)	
Heart failure, *n* (%)	4 (2.8%)	3 (2.1%)	
Pulmonary embolism, *n* (%)	0 (0%)	2 (1.4%)	
*Key median time (IQR), min*			
Time to CPR by EMS, min	8 [6–11]	8 [6–12]	0.26
Time to ALS arrival, min	11 [8–17]	11 [8–16]	0.53
Time to airway established, min	13 [10–17]	12 [10–16]	0.66
Time to randomization, min	15 [12–20]	14 [11–19]	0.15

**Table 2 brainsci-12-01374-t002:** Post-randomization characteristics and measures, according to the trial arm assignment. Data are reported as median [IQRs] or count (%). CABG = coronary artery bypass; IABP = intra-aortic balloon counter-pulsation; ED = emergency department; EMS = emergency medical system; LBBB = left bundle branch block. “New prehospital cardiac arrest” indicated re-arrest while not yet at the hospital.

	Intervention (*n* = 149)	Control (*n* = 142)	*p-*Value
*Prehospital characteristics*			
Adrenaline, median [IQR], mg	5 [3–7]	4 [2–7]	0.12
Amiodarone, median [IQR], mg	300 [300–300]	300 [300–300]	0.4
Duration CPR by EMS, median [IQR], min	18 [7–30]	13 [6–21]	0.01
Ongoing CPR to hospital, No./total (%)	7 (4.7%)	11 (7.7%)	0.72
New prehospital cardiac arrest, No./total (%)	15 (10.0%)	14 (10%)	0.96
Time to start of EMS cooling, median [IQR], min	17 [14–23]	-	-
Time to ROSC, median [IQR], min	29 [30–35]	25 [19–35]	0.98
Tympanic temperature at ROSC—°C, median [IQR]	35.8 [34.8–36.4]	36 [35.5–36.5]	0.06
Time to hospital arrival, median [IQR], min	51 [43–63]	54 [40–64]	0.98
*Characteristics at hospital admission*			
Tympanic temperature at ED, median [IQR]	34.8 [34.2–35.5]	35.8 [35.4–36.2]	0.002
Glasgow Coma Scale, median [IQR]	3 [3–3]	3 [3,4]	0.25
PaCO_2_, median [IQR], mmHg	33 [27–39]	34 [26–41]	0.66
Arterial pH—value, median [IQR]	7.15 [6.98–7.24]	7.17 [7.03–7.29]	0.06
Base excess, median [IQR], mmol/l	−14 [−20–−9]	−12 [−16–−10]	0.14
Lactate, median [IQR], mmol/l	10.2 [7.7–14.4]	10.3 [7.4–13.9]	0.82
Heart rate, median [IQR], mmol/l	87 [72–103]	86 [68–102]	1.00
Systolic blood pressure, median [IQR], mmHg	115 [92–143]	113 [98–133]	0.68
Mean arterial pressure, median [IQR], mmHg	78 [63–96]	79 [68–94]	0.78
SpO_2_, median [IQR], mmHg	98 [93–99]	98 [95–99]	0.25
Spontaneous breathing, No./total (%)	33 (22.1%)	35 (24.6%)	0.80
ST-elevation/new LBBB on ECG—No./total (%)	39 (26.2%)	32 (22.5%)	0.77
ST-depression >1 mm on ECG—No./total (%)	26 (17.4%)	35 (24.6%)	0.045
*Revascularization and circulatory support during ICU stay, n (%)*			
Angiography after hospital admission	72 (48.3%)	65 (45.8%)	0.96
Angiography during ICU stay	10 (6.7%)	8 (5.6%)	0.72
Angiography after ICU stay	4 (2.6%)	4 (2.8%)	0.67
PCI performed	50 (33.6%)	41 (28.9%)	0.52
CABG performed	5 (3.4%)	1 (0.7%)	0.12
IABP performed	5 (3.4%)	6 (4.2%)	0.70

**Table 3 brainsci-12-01374-t003:** Primary outcome in the study population, which are presented as odds ratio (OR) and 95% confidence intervals (CI). Data are presented in all patients and in the subgroup of patients with shockable (ventricular fibrillation or pulseless ventricular tachycardia) and non-shockable (asystole and pulseless electric activity) initial rhythm. CPC = Cerebral Performance Category.

Primary Outcome	Intervention	Control	Difference (95% CI)	Odds Ratio (95% CI)	*p*-Value
Survival with CPC 1–2 at 90 days, *n* (%)					
*All patients*	56/149 (37.6%)	45/142 (31.7%)	5.9 [−5.2–16.8]	1.19 [0.86–1.63]	0.29
*Shockable rhythm*	48/83 (57.8%)	35/78 (44.9%)	12.9 [−2.4–28.3]	1.29 [0.95–1.75]	0.10
*Non-shockable rhythm*	8/66 (12.1%)	10/64 (15.6%)	−3.3 [−15.3–8.6]	0.79 [0.33–1.87]	0.58

**Table 4 brainsci-12-01374-t004:** Secondary outcomes in the population, which are presented as odds ratio (OR) and 95% confidence intervals (CI). Data are presented in all patients and in the subgroup of patients with shockable (ventricular fibrillation or pulseless ventricular tachycardia) and non-shockable (asystole and pulseless electric activity) initial rhythm. CPC = Cerebral Performance Category.

Secondary Outcome	Intervention	Control	Difference (95% CI)	Odds Ratio (95% CI)	*p*-Value
Overall survival at 90 days, no (%)					
*All patients*	60/149 (40.3%)	52/142 (36.6%)	3.7 [−7.5–14.8]	1.10 [0.82–1.47]	0.09
*Shockable rhythm*	51/83 (61.4%)	41/78 (52.6%)	8.8 [−6.4–24.1]	1.17 [0.89–1.53]	0.25
*Non-shockable rhythm*	9/66 (13.6%)	11/64 (17.1%)	−3.5 [−15.8–9.1]	0.81 [0.36–1.81]	0.36
Survival with CPC 1 at 90 days, no (%)					
*All patients*	50/149 (33.6%)	35/142 (24.6%)	9.0 [−1.5–19.3]	1.36 [0.94–1.96]	0.11
*Shockable rhythm*	45/83 (54.2%)	27/78 (34.6%)	19.6 [4.6–34.6]	1.57 [1.09–2.25]	0.01
*Non-shockable rhythm*	5/66 (7.6%)	8/64 (12.5%)	−5.0 [−15.2–5.6]	0.62 [0.21–1.78]	0.60

## Data Availability

Data are available upon request to the authors.

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
