# Peer review of "Intra-Arrest Therapeutic Hypothermia and Neurologic Outcome in Patients Admitted after Out-of-Hospital Cardiac Arrest: A Post Hoc Analysis of the Princess Trial"

_brainsci, 2022, doi:10.3390/brainsci12101374_

Round 1

Reviewer 1 Report

I read with interest this post hoc analysis of the PRINCESS RCT. This is a well-written manuscript.

The primary hypothesis is clearly described and the results adequately reported with respect to primary endpoint. The authors seem to emphasize that VF/VT patients admitted to ICU might benefit from intranasal cooling with respect to full neurological recovery (higher proportion of CPC1 patients vs. controls).

In this regard, I suggest to report distribution of CPC scores in intranasal cooling vs control patients in a figure for all patients, for patients with VF/VT, and for patients with non shockable rhythm. This type of figure could greatly improve the readability of the study.

Author Response

  1. I read with interest this post hoc analysis of the PRINCESS RCT. This is a well-written manuscript. The primary hypothesis is clearly described and the results adequately reported with respect to primary endpoint. The authors seem to emphasize that VF/VT patients admitted to ICU might benefit from intranasal cooling with respect to full neurological recovery (higher proportion of CPC1 patients vs. controls). In this regard, I suggest to report distribution of CPC scores in intranasal cooling vs control patients in a figure for all patients, for patients with VF/VT, and for patients with non-shockable rhythm. This type of figure could greatly improve the readability of the study.

Authors’ response : This has been added, as requested.

Reviewer 2 Report

Dear authors,

Thank you for letting me review your manuscript, which was a lot of work, congratulations on this.

I read it with eager interest, and these are my comments:

Abstract:

Succinct and informative. PICO was answered. The study aim is very short, even though you explained it in the methods. It might be in the reader's interest to rewrite your study aim.

Line 21: secondary 'analysis' - word missing, and patients admitted to (not at) the hospital.

Introduction:

The research question was elaborated well enough; relevant literature was cited. Line 44: resulted 'in'

Line 58: 'trend' towards...

Line 60: unconscious OHCA - I wouldn't know of any conscious OHCA patients. In general, you could improve the manuscript's writing and grammar.

Materials and Methods:

You described the methods of the PRINCESS trial detailed and added appropriate procedures for this analysis. What was the targeted temperature of the prehospital cooling? How cold were the mixture of air/oxygen and liquid coolant? Jumping to the results, the prehospital cooling phase did not seem to be very quick. An explanation might be helpful.

The statistical analysis plan was robust and reasonable.

However, I missed an adjustment for confounders. You added this to your limitations but did not mention why you skipped that.

Again, I found some obvious mistakes.

Line 72: ...trial was 'a' randomised...

Line 87: ...trial 'were': ...

Line 106: 'In' case of ROSC... hospital 'under' CPR..

Line 131: Secondary outcomes were...

Results:

Neat baseline characteristics table, same remains true for table 2. I guess you did not gather these data, but in my opinion, approximate delay until the start of CPR would have been interesting. Even though only witnessed CA patients were enrolled, the time until the onset of CPR might change the neurologic outcome.

What exactly is meant by 'New prehospital cardiac arrest'? In my understanding, you included pts with OHCA. Did you mean that an ambulance crew was tasked with an emergency that evolved to CPR while on scene? I found it very interesting that your PaCO2 median values were so low (also dependent on the airway device). This is unexpected, based on reports from other studies.

Discussion:

Honest and valuable discussion, packed with appropriate citations.

The section starting at line 244 was a bit more nebulous to me - what is your intention in writing this? Does it add anything useful that is in direct context to your analysis? In my opinion, this section can be shortened and rewritten.

Conclusion:

A good summary of the main findings and primary outcome. Nothing to add here.

References:

There are a few writing errors/symbols replacing special letters: References 1, 2, 3, 19, 21, 

In summary, the novelty of this secondary post hoc analysis is limited but might be helpful for further prospective studies, as stated by you. Please improve your writing and grammar throughout the manuscript; proofreading by a native should be considered.

Otherwise, I recommend a minor revision as your design is robust and methodologically well conducted, except for the few improvements I explained.

Author Response

  1. Dear authors, Thank you for letting me review your manuscript, which was a lot of work, congratulations on this. I read it with eager interest, and these are my comments: Abstract: Succinct and informative. PICO was answered. The study aim is very short, even though you explained it in the methods. It might be in the reader's interest to rewrite your study aim.

Authors’ response: We understand the reviewer; however, we had to reduce the text because of length limitations of the abstract.

  1. Line 21: secondary 'analysis' - word missing, and patients admitted to(not at) the hospital.

Authors’ response: The text has been corrected, accordingly.

  1. Introduction: The research question was elaborated well enough; relevant literature was cited. Line 44: resulted 'in'

Authors’ response: The text has been corrected, accordingly.

  1. Line 58: 'trend' towards...

Authors’ response: The text has been corrected, accordingly.

  1. Line 60: unconscious OHCA - I wouldn't know of any conscious OHCA patients. In general, you could improve the manuscript's writing and grammar.

Authors’ response: We thank the reviewer for the important comment. There are some patients with very short no-flow or low-flow time who might rapidly recover consciousness on site. This is why we preferred to keep the definition  of “unconscious” patients. The manuscript has also been edited, accordingly.

  1. Materials and Methods: You described the methods of the PRINCESS trial detailed and added appropriate procedures for this analysis. What was the targeted temperature of the prehospital cooling? How cold were the mixture of air/oxygen and liquid coolant? Jumping to the results, the prehospital cooling phase did not seem to be very quick. An explanation might be helpful.

Authors’ response: We understand the criticism from the reviewer. However, we referred to the main publication, which reported all details about interventions.

  1. The statistical analysis plan was robust and reasonable. However, I missed an adjustment for confounders. You added this to your limitations but did not mention why you skipped that.

Authors’ response: Adjusted odds ratios for the primary outcomes have also been calculated and reported, accordingly.

  1. Again, I found some obvious mistakes. Line 72: ...trial was 'a' randomised... - Line 87: ...trial 'were': ... - Line 106: 'In' case of ROSC... hospital 'under' CPR.. - Line 131: Secondary outcomes were...

Authors’ response: We apologize for the typos, which have been corrected.

  1. Results: Neat baseline characteristics table, same remains true for table 2. I guess you did not gather these data, but in my opinion, approximate delay until the start of CPR would have been interesting. Even though only witnessed CA patients were enrolled, the time until the onset of CPR might change the neurologic outcome.

Authors’ response: Time to CPR from the EMS was reported in Table 1, accordingly.

  1. What exactly is meant by 'New prehospital cardiac arrest'? In my understanding, you included pts with OHCA. Did you mean that an ambulance crew was tasked with an emergency that evolved to CPR while on scene? I found it very interesting that your PaCO2 median values were so low (also dependent on the airway device). This is unexpected, based on reports from other studies.

Authors’ response: “New prehospital cardiac arrest” indicated re-arrest while still not at the hospital. A separate analysis on airway management is under way, accordingly. PaCO2 values have been checked and corrected, accordingly.

  1. Discussion: Honest and valuable discussion, packed with appropriate citations. The section starting at line 244 was a bit more nebulous to me - what is your intention in writing this? Does it add anything useful that is in direct context to your analysis? In my opinion, this section can be shortened and rewritten.

Authors’ response: We do think this section is relevant to help the reader understanding the importance of this analysis, including only admitted patients.

  1. Conclusion: A good summary of the main findings and primary outcome. Nothing to add here.

Authors’ response: Thanks for the kind comment.

  1. References: There are a few writing errors/symbols replacing special letters: References 1, 2, 3, 19, 21.

Authors’ response: These have been corrected, as requested.

Reviewer 3 Report

With a great interest I read the work of Macchini et al. on the intra-arrest therapeutic hypothermia and neurologic outcome in patients admitted after out-of-hospital cardiac arrest. Authors should be congratulated for a great work done.

I would recommend the following adaptations of the work:

Abstract is missing flow. Please revise. 

Section 2,3: Please provide short explanation of the intervention, even if citation provided. What kind of central cooling system was used? Please provide information. 

Section 2.4: Was data on ECMO use collected? Report on other collected variables is missing?

Line 148: Please use the statistically correct numbers presentation: 291 of 677 is 42.98, which is 43%. Please corret the whole work, as this was often the problem. 

Table 1 and all tables: P value should be always presented with 3 decimals i.e. 0.003. Please correct all tables and in text. 

Percents in all tables are not correct. i.e. 70/149 is not 51%. It is very complex to read work with wrongly presented results. 

Height in intervention group is for sure not 117cm.

Line 158, percents again not correct.

Table 2: New prehospital arrest: What does this variable mean? I.e. In intervention 15 patients (again wrong percents) had new prehospital arrest. Did the rest had it for the second time?

How do you explain pCO2 with median 13 and 14 mmHg, if the ROSC is there?

Please provide pH value with 3 decimals, i.e. 7.125. in whole text.

IABP is not performed. Please revise.

Table 3 and 4 are with wrong percents. Do you present OR or RR? Please use unique shortcuts and measurements. 

Discussion is missing good scientific writting and flow. 

Limitations should be extended in the last paragraph. The second paragraph of discussion is rather limitations than discussion. 

Discussion is missing good comparison and discussion of results, it is rather focused on the limitations. Please revise the discussion.

Please provide STROBE statement in the supplementary material. 

Author Response

  1. With a great interest I read the work of Macchini et al. on the intra-arrest therapeutic hypothermia and neurologic outcome in patients admitted after out-of-hospital cardiac arrest. Authors should be congratulated for a great work done.

Authors’ response: Thanks for the nice comment.

  1. Abstract is missing flow. Please revise. 

Authors’ response: We have structured the abstract to fit with the length request, accordingly.

  1. Section 2,3: Please provide short explanation of the intervention, even if citation provided. What kind of central cooling system was used? Please provide information. 

Authors’ response: We understand the comment of the reviewer. We have referred to the main publication to avoid excessive overlap with the original publication. As this is a secondary post hoc analysis, we do not fell this would be a great limitation for the readability of the manuscript.

  1. Section 2.4: Was data on ECMO use collected? Report on other collected variables is missing?

Authors’ response: We reported only data on collected variables, accordingly.

  1. Line 148: Please use the statistically correct numbers presentation: 291 of 677 is 42.98, which is 43%. Please correct the whole work, as this was often the problem. 

This has been corrected, accordingly.

  1. Table 1 and all tables: P value should be always presented with 3 decimals i.e. 0.003. Please correct all tables and in text. 

Authors’ response: We kindly disagree with the reviewer; two decimals are quite sufficient for data interpretation considering the number of included patients.

  1. Percents in all tables are not correct. i.e. 70/149 is not 51%. It is very complex to read work with wrongly presented results. 

Authors’ response: Numbers have been checked, accordingly.

  1. Height in intervention group is for sure not 117cm.

Authors’ response: We apologize for the typo, which has been corrected.

  1. Line 158, percents again not correct.

Authors’ response: Numbers have been checked, accordingly.

  1. Table 2: New prehospital arrest: What does this variable mean? I.e. In intervention 15 patients (again wrong percents) had new prehospital arrest. Did the rest had it for the second time?

Authors’ response: “New prehospital cardiac arrest” indicated re-arrest while still not at the hospital. A separate analysis on airway management is under way, accordingly.

  1. How do you explain pCO2 with median 13 and 14 mmHg, if the ROSC is there?

Authors’ response: Numbers have been checked and corrected, accordingly.

  1. Please provide pH value with 3 decimals, i.e. 7.125. in whole text.

Authors’ response: pH data were collected with two decimals.

  1. IABP is not performed. Please revise.

Authors’ response: It has been performed in few patients – data are available in Table 2.

  1. Table 3 and 4 are with wrong percents. Do you present OR or RR? Please use unique shortcuts and measurements. 

Authors’ response: OR were presented for primary and secondary outcomes analysis (as discussed into the Methods section). Risk ratio has been used for exploratory analyses. Numbers have been checked and corrected, accordingly.

  1. Discussion is missing good scientific writing and flow. Limitations should be extended in the last paragraph. The second paragraph of discussion is rather limitations than discussion. Discussion is missing good comparison and discussion of results, it is rather focused on the limitations. Please revise the discussion. Please provide STROBE statement in the supplementary material. 

Authors’ response: We kindly disagree with the reviewer. The role of TNEC for intra-arrest hypothermia has been extensively discussed in PRINCE, PRINCESS and the individual patient pooled data meta-analysis. As such, we have tried to focus on additional issues (to avoid overlap with previous publications) and highlight the clinical relevance to focus on only admitted patients. We have provided a statement on STOBE criteria in the text. Limitations have been revised, accordingly.

Round 2

Reviewer 3 Report

Dear authors. Thank you very much for your revision. However, my humble opinion is that the review is there to improve your work and reduce potential mistakes. My goal was not the discussion if certain suggestions should be or should be not incorporated (change of wrong calculated percent, reporting of p-value in form of <0.01, or discussion improvement of missing relevant literature). Without incorporation of these relevant suggestions, I would rather disagree with the publication of your work.

I would urge on you to incorporate following changes:

-       2. Abstract is still missing flow and nice scientific stile. Please revise. Abstract is the face of your work, and should be perfect presentation of your results. The first sentence is rather surprising and continues to aim oft he study without any connection.

-       3. I disagree with the authors opinion. Please provide information on the intervention used.

-       6. I disagree with the authors opinion. Reporting on „< 0.01“ is not acceptable. It should be <0.001. Please provide statistical reporting as earlier suggested.

-       5. And 7. I disagree with the authors opinion. The correction is not performed. Line 179: Percent are incorrect. Line 198 Percent are incorrect. Table 3 and 4. Percent are incorrect. This continues further on in paper. Please correct this before next submission.

-       11. How did u measure PaCO2, median (IQR), mmHg prehospitaly?

-       pH value of arterial blood?

-       Methods: Please add in methods section information on data collection for table 2

-       12. I disagree with the authors opinion. There is extreme difference in pH of 7.129 and 7.115, which would be rounded into 7.13 and 7.12. State this as limitation.

-       15. Recommendation not implemented. Discussion is still missing good scientific writing and flow. It has been changed with two new sentences and 8 separate words. Flow is still not acceptable. Scientific writing style is not acceptable. Discussion is still missing good comparison and discussion of results, it is rather focused on the limitations. Please revise the discussion.

-       15. Recommendation not implemented. Limitations have to be extended in the last paragraph. Authors provided minimum extension of limitations, still missing serious limitations of own study.

-       STROBE statement – supplementary not provided
